# Dietary supplementation with copper nanoparticles influences the markers of oxidative stress and modulates vasodilation of thoracic arteries in young Wistar rats

Michał Majewski[1]*, Bernadetta Lis[2], Beata Olas[2], Katarzyna Ognik[3], Jerzy Juśkiewicz[4]

**1** Department of Pharmacology and Toxicology, Faculty of Medicine, University of Warmia and Mazury in Olsztyn, Poland, **2** Department of General Biochemistry, Faculty of Biology and Environmental Protection, University of Łódź, Poland, **3** Department of Biochemistry and Toxicology, Faculty of Biology, Animal Sciences and Bioeconomy, University of Life Sciences, Lublin, Poland, **4** Division of Food Science, Institute of Animal Reproduction and Food Research of the Polish Academy of Sciences, Olsztyn, Poland

* michal.majewski@uwm.edu.pl

**Data Availability Statement:** All relevant data are within the manuscript and its Supporting Information files.

## Abstract

We aimed to study the physiological effects of diet supplemented with copper (Cu) nanoparticles (NPs). During the eight weeks of the experiment, young Wistar rats (at seven weeks of age, $n = 9$) were supplemented with 6.5 mg of Cu either as NPs or carbonate salt ($Cu_{6.5}$). A diet that was not supplemented with Cu served as a negative control ($Cu_0$). The impact of nano Cu supplementation on lipid (reflected as thiobarbituric acid reactive substances– TBARS) and protein peroxidation (thiol and carbonyl groups) in blood plasma as well as the influence on the vasodilatory mechanism(s) of isolated rat thoracic arteries were studied. Supplementation with Cu enhanced lipid peroxidation (TBARS) in $NP_{6.5}$ (x2.4) and in $Cu_{6.5}$ (x1.9) compared to the negative control. Significant increase in TBARS was also observed in $NP_{6.5}$ (x1.3) compared to the $Cu_{6.5}$ group. The level of thiol groups increased in $NP_{6.5}$ (x1.6) compared to $Cu_{6.5}$. Meanwhile, significant (x0.6) decrease was observed in the $Cu_{6.5}$ group compared to the negative control. Another marker of protein oxidation, carbonyl groups increased in $NP_{6.5}$ (x1.4) and $Cu_{6.5}$ (x2.3) compared to the negative control. However significant difference (x0.6) was observed between $NP_{6.5}$ and $Cu_{6.5}$. Arteries from Cu supplemented rats exhibited an enhanced vasodilation to gasotransmitters: nitric oxide (NO) and carbon monoxide (CO). An enhanced vasodilation to NO was reflected in the increased response to acetylcholine (ACh) and calcium ionophore A23187. The observed responses to ACh and CO releasing molecule (CORM-2) were more pronounced in $NP_{6.5}$. The activator of cGMP-dependent protein kinases (8-bromo-cGMP) induced similar vasodilation of thoracic arteries in $NP_{6.5}$ and $Cu_0$ groups, while an increased response was observed in the $Cu_{6.5}$ group. Preincubation with the inducible nitric oxide (iNOS) synthase inhibitor– 1400W, decreased the ACh-induced vasodilation in $NP_{6.5}$, exclusively. Meanwhile the eicosanoid metabolite of arachidonic acid (20-HETE) synthesis inhibitor–HET0016, enhanced vasodilation of arteries from $Cu_0$ group. In conclusion, this study demonstrates that supplementation with nano Cu influences oxidative stress, which further has modified the vascular response.

**Funding:** The work was supported in part by the Medical University of Olsztyn, Poland, Statutory funding, grant No.61.610.001-110 (MM). The funder had no role in study design, data collection and analysis, decision to publish, or preparation of the manuscript.

**Competing interests:** The authors have declared that no competing interests exist.

## Introduction

Copper (Cu) is one of the most important microelements involved in energy metabolism, antioxidant defense, and the synthesis of neurotransmitters and neuropeptides. Cu is involved in tryptophan metabolism by regulating the activity of enzymes in the kynurenine pathway [1], which can generate toxic products when dysregulated [2]. Moreover, Cu is able to modulate systemic inflammation by inducing arachidonic acid conversion and prostanoid synthesis [3].

Recently, nanoparticles (NPs) have emerged as important players in modern medicine. However, NPs tend to exhibit quite different properties, when compared to larger particles of the same element. Once entering circulation, NPs interact with the endothelium and induce nitric oxide (NO) signaling impairment. Oxidative stress and inflammatory response is the mechanism of metal NPs [4,5].

We have previously reported that an oral exposure to nano Cu (6.5 mg Cu/kg of a diet) modulated the antioxidant capacity of blood plasma and vascular response [6]. Modified response to ACh [6] and increased contraction to prostaglandin $F_2$-alpha were observed [7] with no attenuation of endothelin–1-induced contraction [7]. Surprisingly, several studies have reported a beneficial anti-diabetic and cardioprotective role of CuNPs with a decreased production of inflammatory mediators [8,9,10,11,12].

Oxidative modifications of plasma lipids and proteins have been previously reported in various conditions, including cardiovascular disorders, with a great influence on the vascular reactivity [13].

The vascular endothelium plays an important role in maintaining cardiovascular homeostasis by synthesizing and releasing several vasoactive substances, including vasodilator and vasoconstrictor prostanoid, gasotransmitters: nitric oxide (NO) and carbon monoxide (CO), and endothelium-derived hyperpolarizing factors [14].

Endothelial dysfunction is usually associated with a reduction in NO production and/or an increase in NO metabolism. On the other hand, overproduction of NO by the inducible form of NO synthase (iNOS) may contribute to hypotension, cardio-depression and vascular hyporeactivity.

Next to NO, another gasotransmitter, CO, plays an important physiological role in the regulation of vascular tone and inflammation. It is very important that in cardiovascular system, CO does not work often in an isolation, but it may interact with reactive oxygen species (ROS) or reactive nitrogen species (RNS), i.e. NO [15].

Moreover, arachidonic acid metabolites, which are produced through cytochrome P450 (CYP450) enzymes influence cardiovascular homeostasis. 20-Hydroxyeicosatetraenoic acid (20-HETE) is a major biologically active CYP450 metabolite. Human CYP4A11 and CYP1A2 metabolize arachidonic acid to 20-hydroxyecostearonic acid (20-HETE), which is a vasoconstrictor. HET0016 confers anti-oxidative and anti-inflammatory effects in the arteries by disrupting 20-HETE-mediated signaling pathways in the vascular wall [16].

In view of presented data, we aimed to determine the effects of nano Cu supplementation on oxidative stress markers, i.e. lipid peroxidation (reflected as thiobarbituric acid reactive substances–TBARS) and protein oxidation level (thiol and carbonyl groups) in blood plasma. Moreover we aimed to analyze the endothelium-dependent response to: *(i)* analog of cGMP, *(ii)* calcium ionophore, A23187, which is muscarinic receptor independent mechanism, and *(iii)* CO donor–CORM-2. As the mechanism(s) involved in ACh-induced vasodilation of thoracic arteries remain controversial, we aimed to explain this effect with the inducible nitric oxide synthase (iNOS) inhibitor– 1400W, and the 20-HETE synthesis inhibitor– HET0016.

## Materials and methods

### Drugs and reagents

The drugs used were: acetylcholine (ACh) as chloride, noradrenaline (NA) as hydrochloride, A23187, tricarbonyl-dichlororuthenium (II) dimer (CORM-2), 8-bromo-cGMP, 1400W, A23187 (Sigma-Aldrich, St. Louis, MO, USA) and HET0016 (Cayman chemical, Ann Arbor, MI, USA). Stock solutions (10 mM) of these drugs were prepared in distilled water, except for NA which was dissolved in NaCl (0.9%) + ascorbic acid (0.01% w/v) solution. HET0016 was dissolved in ethanol. CORM-2 and 1400W were dissolved in methanol. The maximal solvent concentration in the medium was less than 0.001% (vol/vol). These solutions were maintained at –20°C and appropriate dilutions were made in Krebs-Henseleit solution (KHS in mmol/l: NaCl 115; $CaCl_2$ 2.5; KCl 4.6; $KH_2PO_4$ 1.2; $MgSO_4$ 1.2; $NaHCO_3$ 25; glucose 11.1) on the day of the experiment.

Cu carbonate (purity $\geq$ 99%) was sourced from Poch (Gliwice, Poland). The nano Cu particles (40–60 nm size nanopowder, 12 $m^2$/g) were purchased from Sky Spring Nanomaterials, Inc. (Houston, TX, US), with a purity of 99.9% on a trace metals basis, with a spherical morphology of 0.19 $g/cm^3$ bulk density, and an 8.9 $g/cm^3$ true density. The zeta potential of NPs was determined to be –30.3 mV (phosphate-buffered saline), for more details see Ognik et al [12].

Thiobarbituric acid was purchased from Sigma-Aldrich (St. Louis, MO, USA). All other reagents were of analytical grade and were provided by commercial suppliers.

### Ethical statements

All procedures were approved by the Local Ethics Committee for Animal Experiments in Olsztyn, Poland (*Permission Number*: *65/2017*) according to European Union guidelines (Directive 2010/63/EU for animal experiments) and conform to the *Guide for the Care and Use of Laboratory Animals* published by the US National Institutes of Health (NIH Publications No. 86–26, revised 2014). The 3R rule ("Replacement, Reduction and Refinement") was respected in the study. All surgery was performed under ketamine+xylazine anesthesia, and all efforts were made to minimize animal suffering.

### Experimental protocol

Healthy male albino Wistar rats (Han IGS rat [Crl: WI(Han)]), were housed individually in stainless steel cages under a stable temperature of 21–22°C, a ventilation rate of 20 air changes *per* hour and a relative humidity of 50 ± 10%. During this period, the rats had free access to tap water and were fed *ad libitum*.

At seven weeks of age, rats were randomly divided into three groups of 9 animals each. Experimental rats were supplemented with a standard dose of Cu (6.5 mg/kg of diet) [7,11,17,18] either as Cu carbonate ($Cu_{6.5}$ –the control group) or nano Cu ($NP_{6.5}$). Moreover, a negative control not supplemented with Cu ($Cu_0$) was implemented in this study. Cu as a nano-suspension was prepared in rapeseed oil [19], and the same amount of pure rapeseed oil was added to the other two experimental diets to have equivalent oil content. The experimental diets were modifications of a casein diet for laboratory rodents recommended by the American Institute of Nutrition, and were prepared weekly and then stored at 4°C in hermetic containers [6].

### Experimental procedures in rats and study analysis

Rats were anesthetized by intraperitoneal injection of ketamine+xylazine (100 mg/kg+10 mg/kg of body weight) and killed by decapitation. Immediately after blood collection, samples

were kept in tubes containing heparin + EDTA as an anticoagulant. Samples were centrifuged at 3,000 *g* for 10 min and blood plasma was separated and stored at –80˚C until further analysis [6].

## Oxidative stress markers

**Lipid peroxidation.** Lipid peroxidation was quantified by measuring the concentration of TBARS. Samples of plasma were transferred to an equal volume of cold 15% (v/v) trichloroacetic acid in 0.25M HCl and 0.37M thiobarbituric acid in 0.25M HCl, immersed in a boiling water bath for 10 min, and then centrifuged at 10,000×g for 15 min, 18˚C. Absorbance was measured at 535 nm (the SPECTROstar Nano Microplate Reader—BMG LABTECH Germany). The TBARS concentration was calculated using the molar extinction coefficient ($\varepsilon$ = 156,000 $M^{-1}$ $cm^{-1}$) and was expressed as nmol/mL of plasma [20,21].

**Protein peroxidation markers.** Protein oxidation was quantified by measuring the carbonyl groups and thiol groups in blood plasma. The carbonyl group concentration was calculated using a molar extinction coefficient ($\varepsilon$ = 22,000$M^{-1}cm^{-1}$), and was expressed as nmol carbonyl groups/mg of protein. The detection of protein carbonyls involves derivatization of the carbonyl groups with 2,4-dinitrophenylhydrazine (DNPH), which leads to the formation of a stable 2,4-dinitrophenyl (DNP) hydrazone product [21,22].

The thiol group content was measured spectrophotometrically (the SPECTROstar Nano Microplate Reader- BMG LABTECH Germany) by absorbance at 412 nm with Ellman's reagent: 5,5′-dithio-bis-(2-nitrobenzoic acid). The thiol group concentration was calculated using a molar extinction coefficient ($\varepsilon$ = 13,600$M^{-1}cm^{-1}$). The level of thiol groups was expressed as nmol thiol groups/mg of protein [21,23,24].

**Vascular reactivity studies.** This was done in accordance to our previous study [6]. Briefly, thoracic arteries were carefully dissected out, cleaned of adherent tissue, and cut into 6–8 aortic rings of 3 to 4-mm length, placed in aerated KHS at 4˚C and pH = 7.4. Thoracic rings were suspended under a resting tension of 1 g in 5-mL tissue baths (eight stagnant Graz Tissue Bath System chambers) containing KHS. The solution had been aerated with a carbogen gas (95% oxygen and 5% carbon dioxide), and maintained at 37˚C during the study. Each ring was connected to an isometric force transducer with amplifier (F-30, TAM-A HSE).

The functional integrity of smooth muscles was checked with KCl (75 mM). After a washout period of 60 min, the presence of the vascular endothelium was confirmed by the ability of ACh (10 μM) to induce relaxation (80%) of the aortic rings precontracted with NA (0.1 μM). Next, the rings were rinsed with KHS for 60 min, and then cumulative concentration-response curves (CCRCs) to CORM-2 (1–100 μM), calcium ionophore–A23187 (0.1 nM–10 μM), and 8-bromo-cGMP (0.1–100 μM) were assessed to determine the relaxant responses of endothelium-intact rings that had been precontracted with submaximal concentrations of NA (0.1 μM).

In another set of experiments, ACh-induced vasodilation (0.1 nM–10 μM) was analyzed in the absence and presence of the iNOS inhibitor– 1400W (1 μM, 30 min), and the eicosanoid metabolite of arachidonic acid (20-HETE) synthesis inhibitor–HET0016 (0.1 μM, 30 min). Only one CCRC was performed on each aortic ring.

## Data analysis and statistics

The calculations and graphs were done and analyzed in GraphPad Prism 8.3. Vasodilation induced by ACh, CORM-2, A23187 and 8-bromo-cGMP was represented as a percentage of the initial contraction elicited by NA (0.1 μM).

The maximal response ($E_{max}$, %), the potency (the $pD_2$ values, measured as the negative logarithm of the concentration causing a half-maximum effect), and the area under the curve (AUC) were determined from the $CCRC_S$ by non-linear regression analysis.

The normality and homogeneity of variance was tested for all data. The group comparison was performed by one- or two-way analysis of variance (ANOVA) followed by a *post-hoc Tukey's* or *Dunnett's T3* test, where appropriate.

The results included in Table 1 are expressed as mean ± SEM and median (with Q1 and Q3), whereas Fig 1 presents results as mean ± SD from $n = 9$ rats *per* each group. Q1 and Q3 are the $25^{th}$ and $75^{th}$ percentiles. N varies among bioassays due to limitations on sample volume collected from animal subjects and/or data outlier detection by the *Grubbs'* test. A value of $p \leq 0.05$ was considered to be statistically significant.

## Results

### Blood biomarkers of oxidative stress

TBARS assay measures malondialdehyde, which is produced by lipid peroxidation of polyunsaturated fatty acids. In comparison to the copper deficient diet, Cu supplementation markedly increased the concentration of TBARS in $NP_{6.5}$ by 2.4-fold and in $Cu_{6.5}$ by 1.9-fold (Table 1, Fig 1A).

However, when comparing two copper supplemented groups, lipid peroxidation was increased in $NP_{6.5}$ by 1.3-fold (Fig 1A).

We have also compared the effects of Cu supplementation on plasma protein oxidation in two assays selected to measure free plasma thiol groups and carbonyl concentrations.

In $Cu_{6.5}$ supplemented rats, both markers of protein oxidation did change; thiol groups decreased to 0.6-fold (Fig 1B), and the plasma protein carbonyl groups increased by 2.3-fold (Fig 1C). Meanwhile, supplementation with $NP_{6.5}$ raised by 1.4-fold carbonyl groups exclusively, compared to $Cu_0$ (Fig 1C).

Moreover, in $NP_{6.5}$ supplemented groups protein thiols increased by 1.6-fold and carbonyl groups decreased to 0.6-fold, compared to $Cu_{6.5}$ (Fig 1B and 1C).

### Vascular reactivity studies

ACh (0.01 nM–10 μM), calcium ionophore A23187 (0.01 nM–10 μM), 8-bromo-cGMP (0.01 μM–1 mM), and CORM-2 (0.1 μM–1 nM) induced a concentration-dependent vasodilation of the isolated rat thoracic rings (Figs 2–6).

**Table 1. Blood plasma biomarkers of lipid and protein oxidation.**

| | TBARS | Thiol groups | Carbonyl groups |
|---|---|---|---|
| **$Cu_{6.5}$** | 0.315 ± 0.022 <br> 0.302 (0.265–0.366) | 2.995 ± 0.207 <br> 3.225 (2.447–3.497) | 1.250 ± 0.090 <br> 1.150 (1.071–1.458) |
| **$NP_{6.5}$** | 0.416 ± 0.018 <br> 0.412 (0.363–0.467) | 4.773 ± 0.743 <br> 4.003 (3.196–6.856) | 0.786 ± 0.075 <br> 0.724 (0.612–0.941) |
| **$Cu_0$** | 0.170 ± 0.019 <br> 0.163 (0.116–0.215) | 5.311 ± 0.591 <br> 5.881 (3.574–6.528) | 0.552 ± 0.041 <br> 0.507 (0.438–0.673) |
| *$p$-value | | | |
| $Cu_{6.5}$ *vs.* $NP_{6.5}$ | 0.0033 | 0.05 | 0.0004 |
| $Cu_{6.5}$ *vs.* $Cu_0$ | 0.0001 | 0.0118 | <0.0001 |
| $NP_{6.5}$ *vs.* $Cu_0$ | <0.0001 | 0.9195 | 0.0299 |
| | *Tukey's test* | *Dunnett's T3 test* | *Tukey's test* |

Values are expressed as the mean ± SEM and median (with Q1 and Q3) from $n = 9$ rats per each group

*$p \leq 0.05$ (one-way ANOVA with multiple comparisons test). N varies among bioassays due to limitations on samples volume collected from animal subjects and/or data outlier detection by the *Grubbs'* test. Q1 and Q3 are $25^{th}$ and $75^{th}$ percentiles.

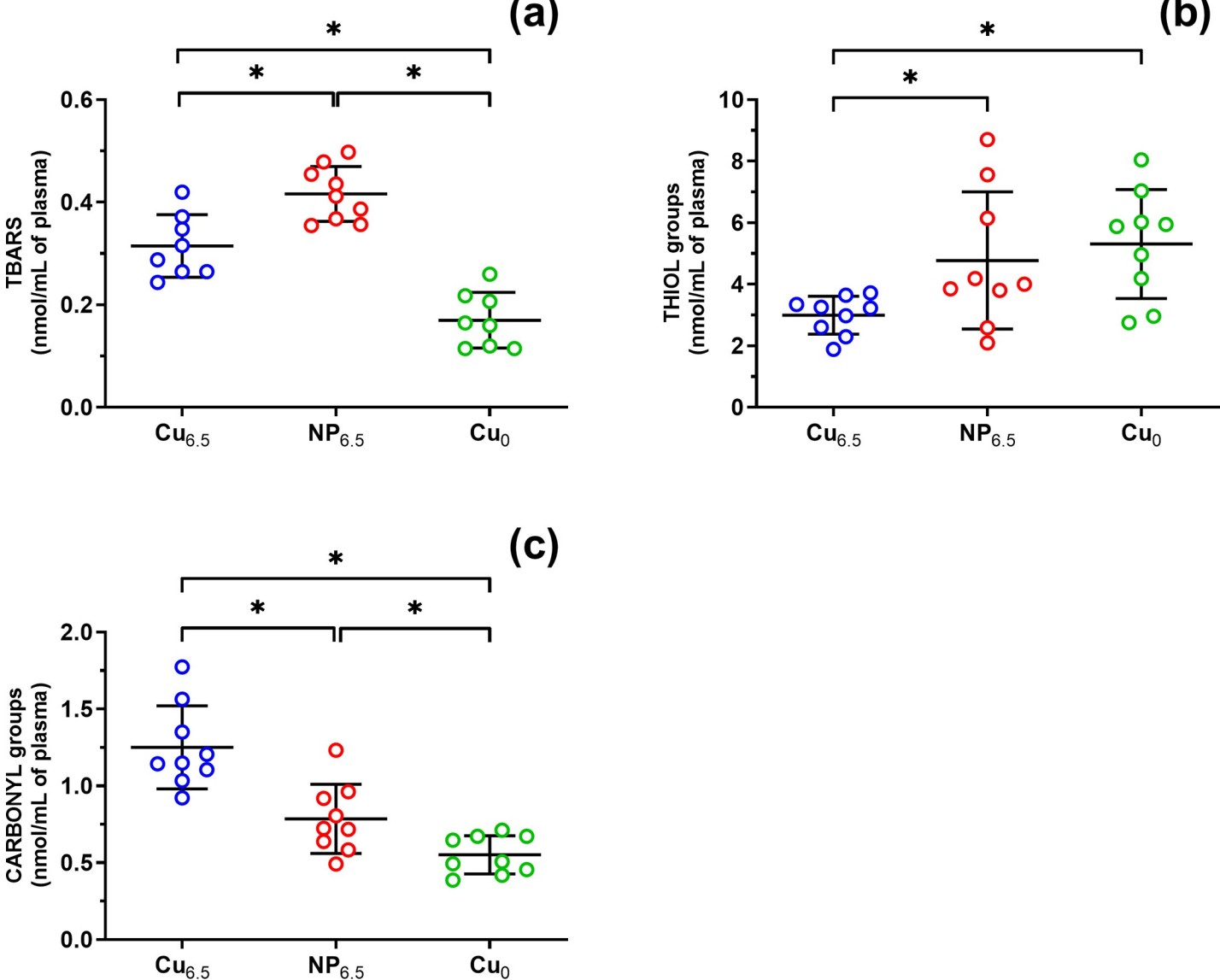

**Fig 1.** Effects of experimental supplementation on (a) lipid peroxidation level–TBARS, (b) the thiol groups and (c) the carbonyl groups. Rats were supplemented with Cu (6.5 mg/kg of diet) either as $Cu_{6.5}$ or $NP_{6.5}$ (40 nm). A diet that was not supplemented with Cu served as a negative control–$Cu_0$. Results are mean ± SD from $n = 9$ rats $^*p \leq 0.05$ (one-way ANOVA with *Tukey's* multiple comparisons test).

The vasodilation induced by ACh was enhanced in the two Cu supplemented groups at the lower concentration range (10–100 nM), and was shifted to the left compared to $Cu_0$ animals (Fig 2A). Significant increase in vasodilation was observed in nano Cu fed rats at lower concentrations, 0.1–1 nM compared to $Cu_{6.5}$.

In the presence of the selective iNOS inhibitor, 1400W (1 µM, 30 min), vasodilation to ACh was enhanced in the $Cu_{6.5}$ group compared to $NP_{6.5}$ and $Cu_0$ (Fig 2B). Meanwhile, in $NP_{6.5}$ exclusively preincubation with 1400W attenuated the vascular response to ACh compared to the control conditions not treated with 1400W (Fig 3B). This was neither observed in the $Cu_{6.5}$ nor in $Cu_0$ group (Fig 3A and 3C).

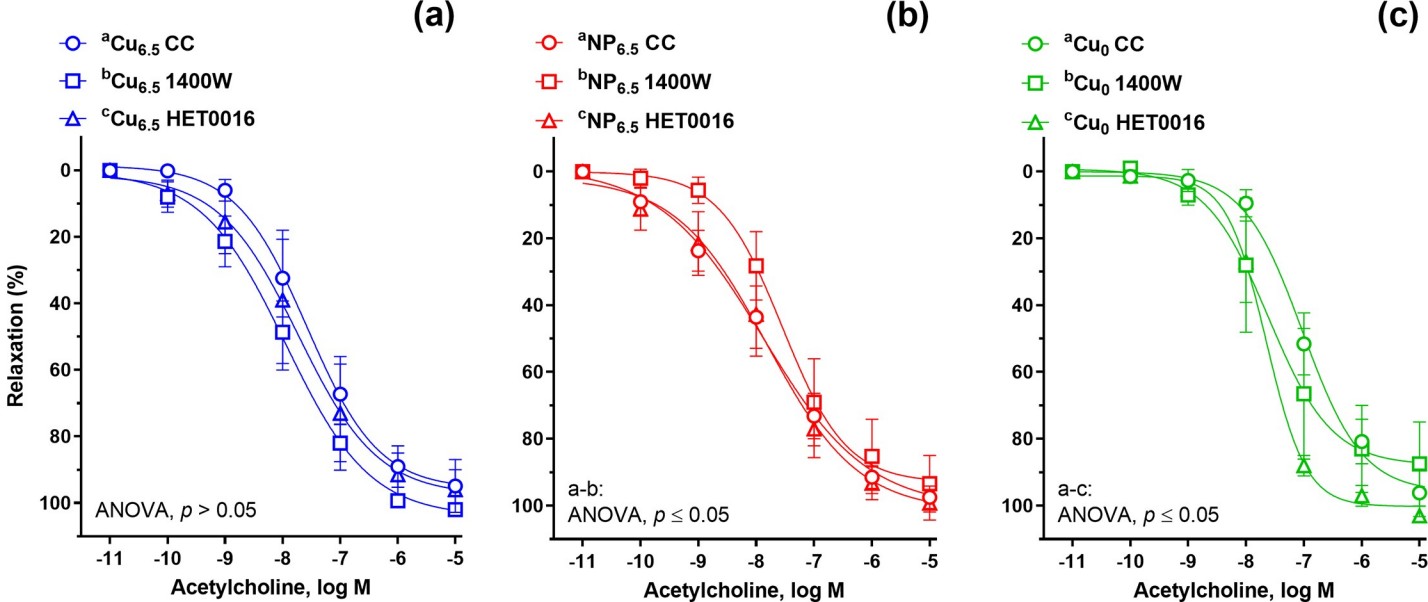

**Fig 2. The cumulative concentration-response curves to acetylcholine (ACh, $10^{-11}$–$10^{-5}$ M) in the isolated rat thoracic arteries.** Rats were supplemented with Cu (6.5 mg/kg of diet) either as $Cu_{6.5}$ or $NP_{6.5}$ (40 nm). A diet that was not supplemented with Cu served as a negative control–$Cu_0$. (a) Control conditions. (b) Isolated rat arteries were pre-incubated with the inducible nitric oxide synthase (iNOS) inhibitor (1400W, 1 μM, 30 min). Results (mean ± SEM, $n = 9$) are expressed as a percentage of inhibition of the contraction induced by noradrenaline (0.1 μM) $^*p \leq 0.05$ (two-way ANOVA with *Tukey's* multiple comparisons test).

Preincubation with the inhibitor of 20-HETE synthesis, HET0016 (0.1 μM, 30 min) resulted in an enhanced vasodilation to ACh in $Cu_0$ rats (Fig 3C). In $Cu_{6.5}$ and $NP_{6.5}$, the vasodilation to ACh, in the presence of HET0016 remained unmodified (Fig 3A and 3B).

Calcium ionophore A23187, induced the concentration-dependent vasodilation which was potentiated in the $NP_{6.5}$ and $Cu_{6.5}$ group at 1–10 nM, compared to $Cu_0$. Moreover, in $NP_{6.5}$ the vasodilation was shifted to the right when compared to Cu6.5 (Fig 4).

The vasodilation induced by the cell-permeable analog of cGMP, 8-bromo-cGMP (0.1–100 μM) was decreased in $Cu_0$ and $NP_{6.5}$ supplemented rats in a similar way, when compared to the $Cu_{6.5}$ group (Fig 5).

The vascular response to CO releasing molecule, CORM-2 was enhanced in Cu supplemented rats at 31.63–100 μM, and to a greater extent in $NP_{6.5}$ (3.16 μM–1 nM) (Fig 6). Significant increase in vasodilation was also observed in $NP_{6.5}$ fed rats at 3.16–10 μM compared to $Cu_{6.5}$, which is similar to the response described for ACh at lower concentration range (0.1–1 nM).

The effects of different drugs used on the $E_{max}$ (%), the $pD_2$ and AUC are summarized in Table 2.

## Discussion

Oxidative stress is a state characterized by the excessive production of reactive oxygen species (ROS), as a result of an imbalance between the intensity of oxidative mechanisms and the counteractive defense systems. Prolonged oxidative stress may lead to cell injury, especially oxidation of proteins, carbohydrates and lipids [25]. The assessment of the intensity of oxidative stress allows to determine the degree of cell damage and related disorders. Oxidative stress is considered to be the main cause of atherosclerosis, diabetes, cataract, asthma, cancer, neurodegenerative and autoimmune diseases [26]. The evaluation of oxidative stress intensity is

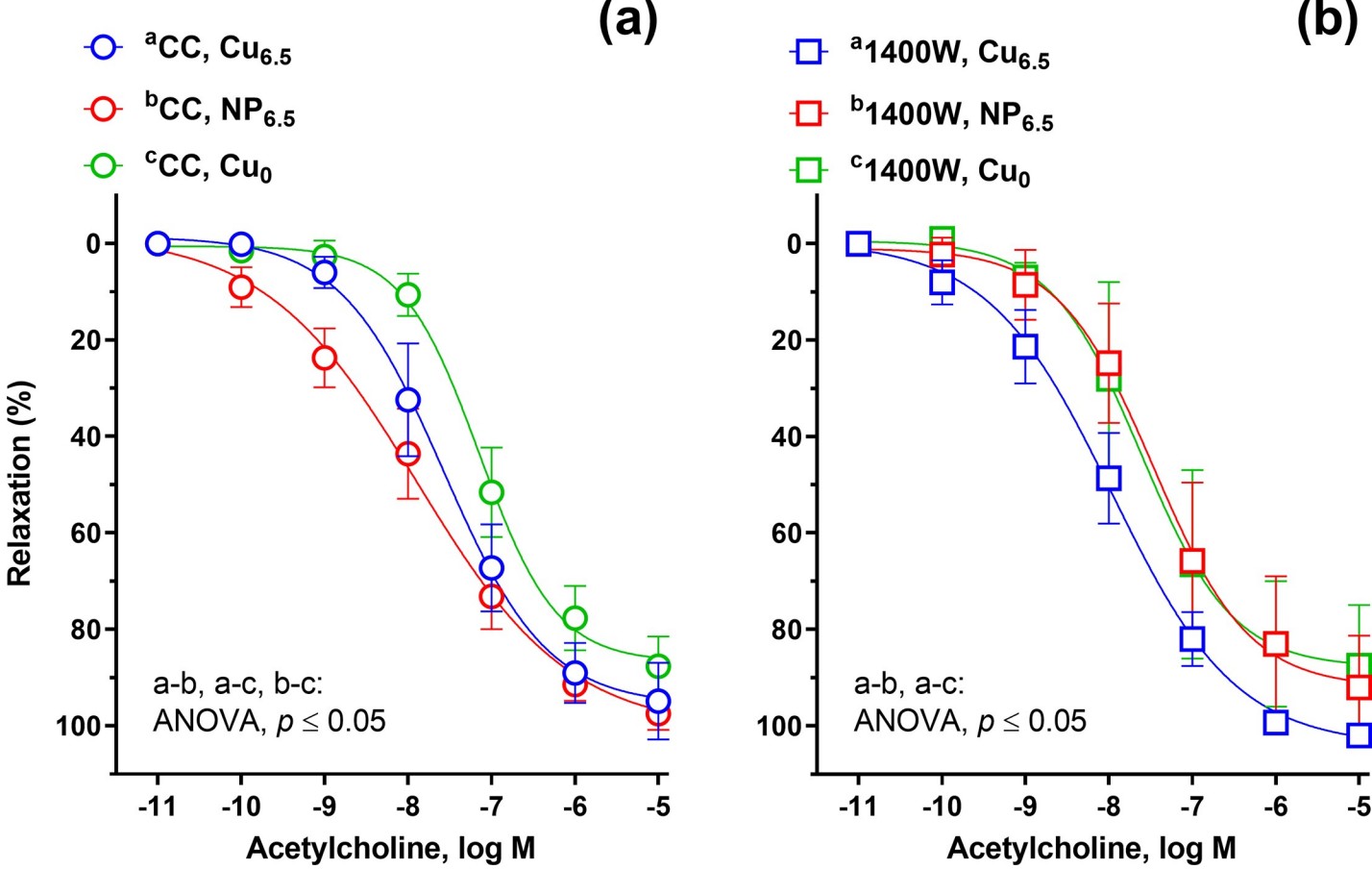

**Fig 3. The cumulative concentration-response curves to acetylcholine (ACh, $10^{-11}$–$10^{-5}$ M) in the isolated rat thoracic arteries.** Rats were supplemented with Cu (6.5 mg/kg of diet) either as (a) $Cu_{6.5}$ or (b) $NP_{6.5}$ (40 nm). (c) A diet that was not supplemented with Cu served as a negative control–$Cu_0$. ACh-induced vasodilation was analyzed in the absence and presence of the iNOS inhibitor (1400W, 1 μM, 30 min), and 20-HETE synthesis inhibitor (HET0016, 0.1 μM, 30 min). Results (mean ± SEM, $n = 9$) are expressed as a percentage of inhibition of the contraction induced by noradrenaline (0.1 μM), $^*p \leq 0.05$ (two-way ANOVA with *Tukey's* multiple comparisons test).

more often based on compounds formed as a result of the reaction of free radicals with components of organism cells, because each of the oxidizing substances causes the formation of characteristic markers, including specific modifications of proteins or reaction by-products [27].

In case of proteins, the markers of their damage are the products of their oxidation, which are formed as a result of oxidative stress. Among the substances which have the ability to oxidize proteins are: hydroperoxides, hypochlorite, bromic acid and reducing metals, e.g. iron and copper. The most important protein oxidation reactions include nitration of cyclic amino acid residues, oxidation of thiol groups (-SH), oxidation of cysteine and methionine residues, or formation of carbonyl derivatives of some amino acids. However, in case of lipids, the markers of their damage are the products of lipid peroxidation, which contribute to changes in the physical properties of cell membranes. Polar peroxides, ketones, aldehydes and hydroxyl polar groups are introduced to phospholipids inside the double lipid layer, which decreases the lipid hydrophobicity of cell membranes and changes the organization of the double lipid layer. As a result, these changes lead to abnormal lipid asymmetry of the membranes.

In our study, three various assays (lipid peroxidation, plasma protein thiol groups and carbonyl determination) have been used to study the antioxidant properties of Cu supplemented and Cu deficient diets. Our results revealed differences in the antioxidant properties between

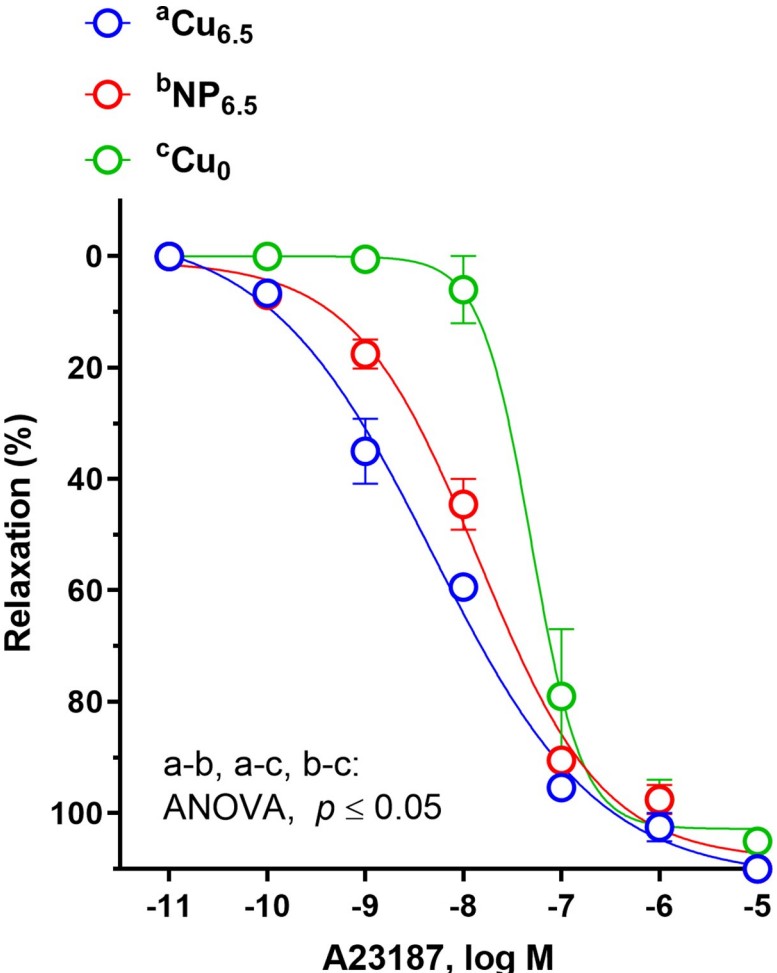

**Fig 4. The cumulative concentration-response curves to calcium ionophore (A23187, $10^{-11}$–$10^{-5}$ M) in the isolated rat thoracic arteries.** Rats were supplemented with Cu (6.5 mg/kg of diet) either as $Cu_{6.5}$ or $NP_{6.5}$ (40 nm). A diet that was not supplemented with Cu served as a negative control–$Cu_0$. Results (mean ± SEM, $n$ = 9) are expressed as a percentage of inhibition of the contraction induced by noradrenaline (0.1 μM), $^*p \leq 0.05$ (two-way ANOVA with *Tukey's* multiple comparisons test).

tested diets. We presented that the oral administration of the typically reported 6.5 mg/kg NP and ionic Cu increased the marker of lipid peroxidation in blood plasma (increased TBARS). Moreover, we also demonstrated that ionic Cu had stronger pro-oxidative effect on proteins than nano Cu (reflected as decreased plasma free thiol groups and increased protein carbonylation/oxidation), see Fig 7.

20-HETE is produced by ω-hydroxylases of the cytochrome P450 in the vascular wall. This increases smooth muscle tone indirectly as a result of its metabolism by COX in endothelial cells [28,29]. Incubation with the 20-HETE synthesis inhibitor, HET0016, did not influence the vasodilation induced by ACh in aortic rings of Cu supplemented rats, discounting the participation of the 20-HETE derivate in this response. Meanwhile, enhanced vasodilation was observed in arteries from copper deprived rats, which suggest the participation of this arachidonic acid metabolite in vascular response in this group, see Fig 7.

ACh-induced NO release by endothelial cells is a muscarinic receptor-mediated effect, whereas the calcium ionophore A23187 response, is a receptor-independent endothelial

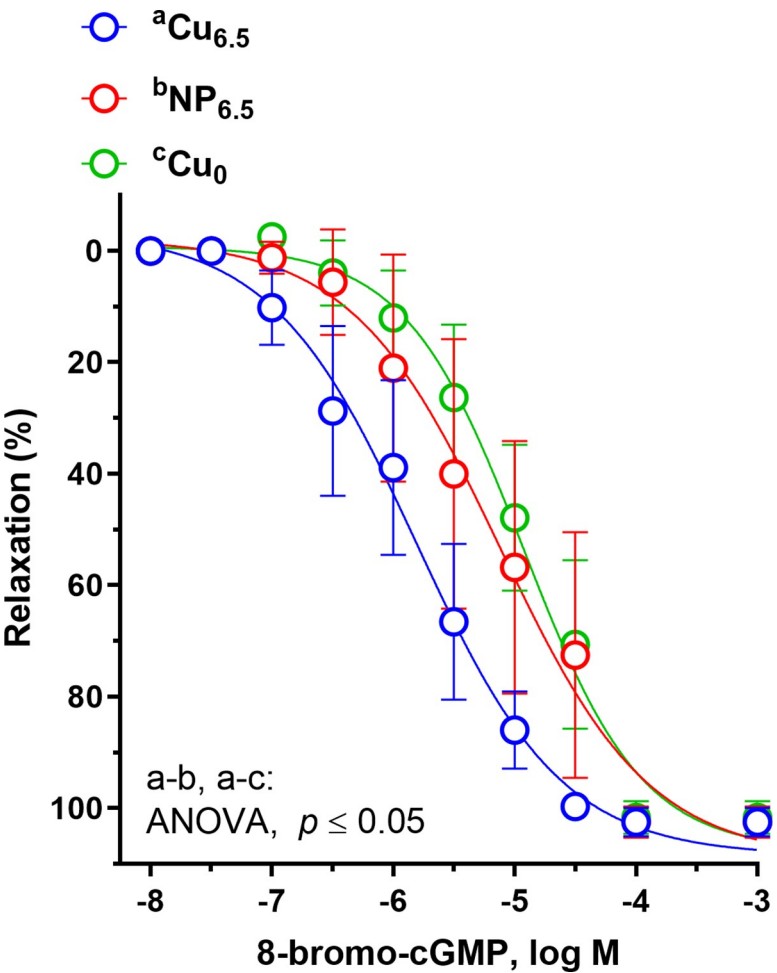

**Fig 5. The cumulative concentration-response curves to a cell-permeable analog of cGMP (8-bromo-cGMP, $10^{-8}$–$10^{-3}$ M) in the isolated rat thoracic arteries.** Rats were supplemented with Cu (6.5 mg/kg of diet) either as $Cu_{6.5}$ or $NP_{6.5}$ (40 nm). A diet that was not supplemented with Cu served as a negative control–$Cu_0$. Results (mean ± SEM, $n = 9$) are expressed as a percentage of inhibition of the contraction induced by noradrenaline (0.1 μM), $^*p \leq 0.05$ (two-way ANOVA with *Tukey's* multiple comparisons test).

mechanism. In our study, A23187 induced a concentration-dependent relaxation of aortic rings from nano Cu and ionic Cu supplemented rats, however to a greater extent in $Cu_{6.5}$ fed rats, which is opposite to the response described for ACh in corresponding groups of rats.

These results suggest that the possible site for nano Cu to interfere with endothelium-dependent relaxation is due to the activation of the muscarinic acetylcholine receptors in the endothelial cells, with subsequent $Ca^{2+}$ release, and it may also occur downstream of the receptors in the bioavailability of endothelial NO.

Reduced relaxation to A23187 was also observed in Cu deficient rats, and has been described previously in streptozotocin- [30] and MSG-induced diabetic rats [31]; as well as in renal hypertensive rats [32].

During the 8 weeks of experimental feeding, the vascular response to cGMP analogue was similar in the nano Cu and Cu deficient group. Meanwhile, in the $Cu_{6.5}$ supplemented rats, the relaxant response was potentiated. This points toward alterations of the vasodilator effect of cGMP-dependent protein kinase (PKG) signaling cascade in $NP_{6.5}$ and $Cu_0$ rats. The observed results might be explained by the fact that soluble guanylate cyclase (sGC) activity and cGMP-

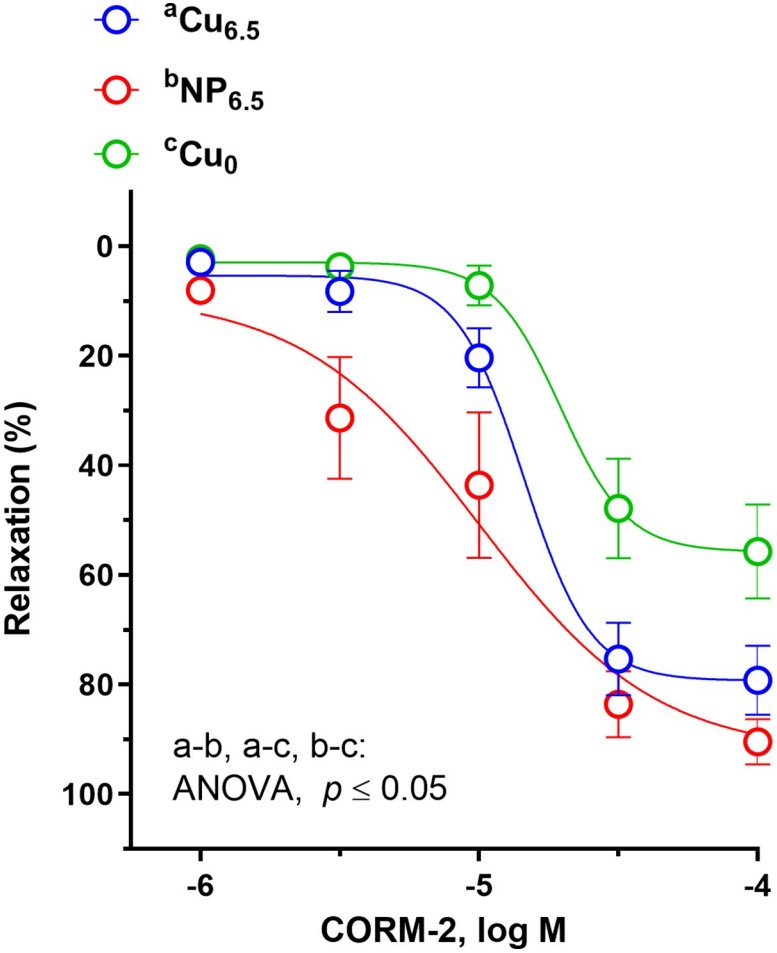

**Fig 6. The cumulative concentration-response curves to carbon monoxide releasing molecule, (CORM-2, $10^{-6}$–$10^{-4}$ M) in the isolated rat thoracic arteries.** Rats were supplemented with Cu (6.5 mg/kg of diet) either as $Cu_{6.5}$ or $NP_{6.5}$ (40 nm). A diet that was not supplemented with Cu served as a negative control–$Cu_0$. Results (mean ± SEM, $n$ = 9) are expressed as a percentage of inhibition of the contraction induced by noradrenaline (0.1 μM), $^*p \leq 0.05$ (two-way ANOVA with *Tukey's* multiple comparisons test).

dependent protein kinase expression have been previously shown to be negatively regulated by proinflammatory mediators [33,34].

Nitric oxide (NO) stimulates sGC activity, and the upregulation of either iNOS or sGC activity represents an early compensatory mechanism to counteract the effects of induced oxidative stress [35].

We observed, that the specific iNOS inhibitor (1400W), reduced the ACh-induced vasodilation in nano Cu rats to the level observed in Cu deficient rats, and that 1400W did not modify this response in $Cu_{6.5}$ fed animals. This may point to the overproduction of NO from iNOS in nano Cu fed rats and may support the above described attenuated vasodilation to 8-bromo-cGMP, as the response to proinflammatory mediators.

Another interesting finding is that the exogenous CO donor, CORM-2, similar to NO and ACh response, enhanced the vasodilation of isolated arteries in nano Cu. However, the vascular response to the CO donor was more significant compared to that of ACh. CO is able to stimulate only four-fold sGC with subsequent cyclic guanosine monophosphate (cGMP) production [36], meanwhile NO stimulates sGC activity 200-fold [37], indicating that CO-induced accumulation of cGMP involves additional cellular components.

**Table 2. Changes in the maximal response (E$_{max}$, %), pD$_2$ parameters and AUC to vasodilators: acetylcholine (ACh), calcium ionophore (A23187), 8-bromo-cGMP and carbon monoxide releasing molecule (CORM-2) in the isolated thoracic rings from experimental Wistar rats.**

| | | Cu$_{6.5}$ [a] | | | NP$_{6.5}$ | | | Cu$_0$ [b] | | |
|---|---|---|---|---|---|---|---|---|---|---|
| | | E$_{max}$ (%) | pD$_2$ | AUC | E$_{max}$ (%) | pD$_2$ | AUC | E$_{max}$ (%) | pD$_2$ | AUC |
| ACh | | 95.72 ± 9.06 [b] | 7.57 ± 0.24 [b] | 337.0 ± 36.61 [b] | 100.9 ± 9.45 [b] | 7.89 ± 0.32 [b] | 387.0 ± 27.23 [a, b] | 86.70 ± 6.06 | 7.14 ± 0.13 | 275.3 ± 28.78 |
| | + 1400W | 104.1 ± 6.06 [b] | 7.96 ± 0.20 | 412.3 ± 17.59 [b] * | 92.76 ± 8.71 [a] | 7.46 ± 0.26 [b] | 322.3 ± 41.04 [a] * | 85.82 ± 11.63 | 7.61 ± 0.36 | 315.9 ± 34.52 * |
| | + HET0016 | 97.59 ± 7.01 | 7.73 ± 0.27 | 370.0 ± 30.54 | 99.98 ± 5.36 | 7.66 ± 0.21 | 394.6 ± 28.87 | 100.1 ± 4.04 * | 7.69 ± 0.11 * | 373.2 ± 16.40 * |
| A23187 | | 111.6 ± 5.71 | 8.31 ± 0.23 [b] | 463.8 ± 8.029 [b] | 108.3 ± 3.66 | 7.84 ± 0.09 [a] | 422.0 ± 5.831 [a, b] | 102.8 ± 3.80 | 7.31 ± 0.11 | 340.5 ± 13.93 |
| 8-bromo-cGMP | | 109.0 ± 15.6 | 5.82 ± 0.26 [b] | 293.1 ± 25.96 [b] | 109.6 ± 18.2 | 5.13 ± 0.36 [a] | 226.7 ± 33.51 [a] | 107.6 ± 9.97 | 4.93 ± 0.20 | 208.0 ± 28.77 |
| CORM-2 | | 79.35 ± 6.94 [b] | 4.84 ± 0.11 | 151.6 ± 11.66 [b] | 95.50 ± 8.30 [a, b] | 5.07 ± 0.26 | 194.3 ± 18.15 [a, b] | 55.73 ± 6.36 | 4.71 ± 0.13 | 99.42 ± 17.68 |

Rats were supplemented for 8 weeks with 6.5 mg Cu/kg of a diet either as copper carbonate (Cu$_{6.5}$) or nano Cu (NP$_{6.5}$, 40 nm). The negative control was not supplemented with Cu–Cu deficient diet (Cu$_0$). Acetylcholine-induced vasodilation was analyzed in the absence and presence of the iNOS inhibitor, 1400W (1 μM, 30 min) and 20-HETE synthesis inhibitor, HET0016 (0.1 μM, 30 min).

* *vs*. ACh control conditions

[a] *vs*. Cu$_{6.5}$

[b] *vs*. Cu$_0$ ($p \leq 0.05$, two-way ANOVA with *Tukey's* multiple comparisons test).

*Abbreviations*: AUC, area under the curve; E$_{max}$, maximal response; pD$_2$, potency.

Another mechanism of CORM-2 involves the activation of Ca$^{2+}$-dependent potassium channels (BK$_{Ca}$). However, previously we did not observe any changes induced by the BK$_{Ca}$ channel opener, NS 1619, between the two Cu supplemented groups [6].

CO has an inhibitory effect on numerous proteins including cytochrome P450 [15], so it may constitute one of the mechanism responsible for the regulation of vascular response. However participation of 20-HETE derivate has been already ruled out. It's worth mentioning that an oral exposure to nano Cu is able to affect drug metabolism by inhibiting the expression of various CYP450 enzymes in the liver [5].

CORM-2 is also a modulator of voltage-gated calcium channels [36], and Cu has been described previously as a modulator of these channels [38]. However, the interaction between CORM-2 and nano Cu, as well as the influence on calcium channels has not been undertaken in this manuscript and requires further studies, see Fig 7 for summary of the results.

It's worth mentioning that both resveratrol and fish oil supplementation had different physiological effects in the nano Cu group of rats compared to ionic Cu [19,39] which have only shown how complex are the observed different effects between these two forms of copper.

## Conclusion

Our results demonstrate that dietary nano Cu and Cu carbonate increase lipid peroxidation (TBARS). However, nano Cu supplementation has different physiological effect towards protein oxidation, which is reflected in an increase in thiols and decrease in carbonyl groups.

Moreover, both nano Cu and carbonate enhance vasodilation to ACh, CORM-2 and A23187, meanwhile supplementation with Cu carbonate enhances vasodilation to 8-bromo-cGMP.

In addition, we provide evidence that although nano Cu may increase NO production through increased iNOS activity, the cGMP/PKG signaling cascade is downregulated, which could diminish the enhanced iNOS activation. Moreover, the 20-HETE-mediated signaling pathway is not involved in a vascular regulation of Cu supplemented rats.

This study demonstrates that supplementation with nano Cu influences oxidative stress which modifies the response of rat thoracic arteries.

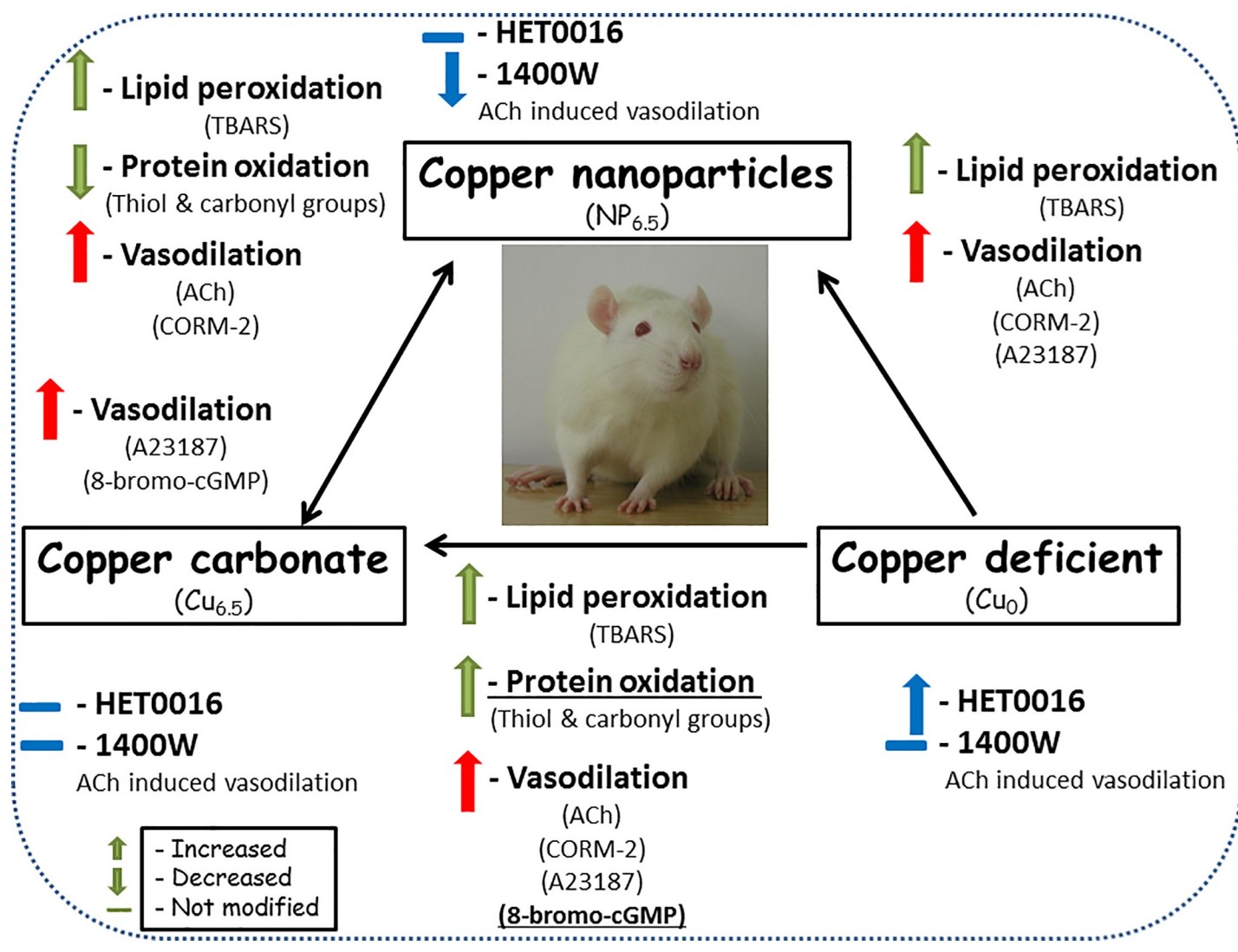

**Fig 7. A summary of the results.**

## Supporting information

**S1 File. Highlights.**
(DOCX)

## Author Contributions

**Conceptualization:** Michał Majewski.

**Data curation:** Michał Majewski.

**Formal analysis:** Michał Majewski.

**Funding acquisition:** Michał Majewski.

**Investigation:** Michał Majewski, Bernadetta Lis, Beata Olas, Katarzyna Ognik, Jerzy Juśkiewicz.

**Methodology:** Michał Majewski.

**Project administration:** Michał Majewski.

**Resources:** Michał Majewski.

**Software:** Michał Majewski.

**Supervision:** Michał Majewski.

**Validation:** Michał Majewski.

**Visualization:** Michał Majewski.

**Writing – original draft:** Michał Majewski.

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
