## [Decision Letter · Decision Letter 0]

11 Dec 2019

PONE-D-19-30444

Copper nanoparticles and carbonate influence the markers of oxidative stress and potentiate vasodilation of thoracic aorta in supplemented Wistar rats

PLOS ONE

Dear Dr Majewski,

Thank you for submitting your manuscript to PLOS ONE. After careful consideration, we feel that it has merit but does not fully meet PLOS ONE’s publication criteria as it currently stands. Therefore, we invite you to submit a revised version of the manuscript that addresses the points raised during the review process. Both reviewers suggest partial rewriting of the manuscript.

We would appreciate receiving your revised manuscript by Jan 25 2020 11:59PM. To enhance the reproducibility of your results, we recommend that if applicable you deposit your laboratory protocols in protocols.io, where a protocol can be assigned its own identifier (DOI) such that it can be cited independently in the future. For instructions see: http://journals.plos.org/plosone/s/submission-guidelines#loc-laboratory-protocols

We look forward to receiving your revised manuscript.

Kind regards,

Michael Bader

Academic Editor

PLOS ONE

Journal Requirements:

**When submitting your revision, we need you to address these additional requirements:**

**Please ensure that your manuscript meets PLOS ONE's style requirements, including those for file naming. The PLOS ONE style templates can be found at http://www.plosone.org/attachments/PLOSOne_formatting_sample_main_body.pdf and http://www.plosone.org/attachments/PLOSOne_formatting_sample_title_authors_affiliations.pdf**To comply with PLOS ONE submissions requirements, please provide methods of sacrifice in the Methods section of your manuscript. We noticed minor instances of text overlap with your previous publication, which need to be addressed:https://doi.org/10.1016/j.pharep.2019.02.007In your revision please ensure you cite all your sources (including your own works), and quote or rephrase any duplicated text outside the methods section. Further consideration is dependent on these concerns being addressed.  Thank you for including your ethics statement: All procedures were approved by the Local Ethics Committee for Animal Experiments (Permission Number: 65/2017) according to European Union guidelines (Directive 2010/63/EU for animal experiments) and conform to the Guide for the Care and Use of Laboratory Animals published by the US National Institutes of Health (NIH Publications No. 86–26, revised 2014). All surgery was performed under sodium pentobarbital anesthesia, and all efforts were made to minimize animal suffering.Please amend your current ethics statement to include the full name of the ethics committee that approved your specific study.For additional information about PLOS ONE submissions requirements for ethics oversight of animal work, please refer to http://journals.plos.org/plosone/s/submission-guidelines#loc-animal-research  Once you have amended this/these statement(s) in the Methods section of the manuscript, please add the same text to the “Ethics Statement” field of the submission form (via “Edit Submission”).

Reviewers' comments:

Reviewer's Responses to Questions

**Comments to the Author**

1. Is the manuscript technically sound, and do the data support the conclusions?

Reviewer #1: No

Reviewer #2: Yes

2. Has the statistical analysis been performed appropriately and rigorously? 

Reviewer #1: No

Reviewer #2: Yes

3. Have the authors made all data underlying the findings in their manuscript fully available?

Reviewer #1: No

Reviewer #2: Yes

4. Is the manuscript presented in an intelligible fashion and written in standard English?

Reviewer #1: No

Reviewer #2: Yes

5. Review Comments to the Author

Reviewer #1: The study presents the influence of Copper nanoparticles and carbonate on oxidative stress and potentiate vasodilation of thoracic aorta in supplemented Wistar rats.

Title itself is not convincing

Conclusion of the study as mentioned in abstract indicates authors need to put extra efforts

English language and grammar is not appropriate. Few sentences itself starting form ‘because’ which is just an example.

How Sample size of the study is derived need to be mentioned

Purpose of the study is not convincing, neither the conclusions derived from the study.

Manuscript should be re-written with specific outcomes.

Reviewer #2: This study investigated the effects of Cu NPs and CuCO3 on oxidative stress and vasodilation of aorta as well as the possible mechanisms. The study is well-designed and performed, and reported some interesting data. I have some comments that may further improve the manuscript.

1. In the title, "supplemented Wistar rats" is very unclear. I recommend the authors to revise the title to make it more clear and straightforward.

2. Please make a conclusion at the end of abstract. The statement "Our findings show that CuNPs possess different biological properties" is not specific enough.

3. The authors stated in the introduction what they are going to do and the reasons to test these endpoints. This is good. However, since the authors wrote two aims and different mechanisms, it is very difficult for the readers to follow all the points. Therefore, I suggest the authors make a figure about the study design. This figure should also include what are the molecules to mediate the effects, and the actions of all chemicals used in this study.

4. Please state where you bought all the chemicals, including the NPs. Also, although the authors bought Cu NPs, it is still necessary to characterize them. At least, the authors should do TEM/SEM, and measure the hydrodynamic size/zeta potential in suspensions. I looked at the authors' previous study (pharmacological reports), but they did not characterize the NPs.

5. The authors compared the effects of Cu NPs and CuCO3. Pleast state (1) why should the comparison be necessary; (2) through the comparison, what conclusions could be made?

6. PLOS authors have the option to publish the peer review history of their article (what does this mean?). If published, this will include your full peer review and any attached files.

Reviewer #1: No

Reviewer #2: Yes: Yi Cao

---

## [Author Response · Author response to Decision Letter 0]

25 Jan 2020

1. Please ensure that your manuscript meets PLOS ONE's style requirements, including those for file naming. The PLOS ONE style templates can be found at http://www.plosone.org/attachments/PLOSOne_formatting_sample_main_body.pdf

and 

I did check and it is in accordance with plos one

http://www.plosone.org/attachments/PLOSOne_formatting_sample_title_authors_affiliations.pdf I did check and it is in accordance with plos one

2. To comply with PLOS ONE submissions requirements, please provide methods of sacrifice in the Methods section of your manuscript. Rats were anesthetized by intraperitoneal injection of ketamine+xylazine [100 mg/kg+10 mg/kg body weight (BW)] and killed by decapitation. Immediately after blood collection, samples were kept in tubes containing heparin + EDTA as an anticoagulant. Samples were centrifuged at 3,000 g for 10 min and blood plasma was separated and stored at –80 °C until further analysis.

3. 

We noticed minor instances of text overlap with your previous publication, which need to be addressed:

https://doi.org/10.1016/j.pharep.2019.02.007

In your revision please ensure you cite all your sources (including your own works), and quote or rephrase any duplicated text outside the methods section. Further consideration is dependent on these concerns being addressed. I did rephrase

4. 

 Thank you for including your ethics statement: All procedures were approved by the Local Ethics Committee for Animal Experiments in Olsztyn, Poland (Permission Number: 65/2017) according to European Union guidelines (Directive 2010/63/EU for animal experiments) and conform to the Guide for the Care and Use of Laboratory Animals published by the US National Institutes of Health (NIH Publications No. 86–26, revised 2014). The 3R rule (“Replacement, Reduction and Refinement”) was respected in the study. All surgery was performed under ketamine+xylazine anesthesia, and all efforts were made to minimize animal suffering.

For additional information about PLOS ONE submissions requirements for ethics oversight of animal work, please refer to http://journals.plos.org/plosone/s/submission-guidelines#loc-animal-research

Reviewers' comments:

Reviewer #1: The study presents the influence of Copper nanoparticles and carbonate on oxidative stress and potentiate vasodilation of thoracic aorta in supplemented Wistar rats.

Title itself is not convincing: Title changed: now it is: Dietary supplementation with copper nanoparticles influences the markers of oxidative stress and modulates vasodilation of thoracic arteries in young Wistar rats

Conclusion of the study as mentioned in abstract indicates authors need to put extra efforts

English language and grammar is not appropriate. Few sentences itself starting form ‘because’ which is just an example. ‘because’ was changed - 

the manuscript was checked by the native English speaker (from US) previously

How Sample size of the study is derived need to be mentioned Sample size - base on my previous studies with n=9, previously the sample size was calculated based on Statistical algorithms developed by Columbia University Medical Center (www.biomath.info).

Purpose of the study is not convincing, neither the conclusions derived from the study.

Manuscript should be re-written with specific outcomes. Manuscript was re-written;

the conclusions was re-written “In conclusion, this study demonstrates that supplementation with nano Cu influences oxidative stress, which further do modify the vascular response”

Reviewer #2: This study investigated the effects of Cu NPs and CuCO3 on oxidative stress and vasodilation of aorta as well as the possible mechanisms. The study is well-designed and performed, and reported some interesting data. I have some comments that may further improve the manuscript.

1. In the title, "supplemented Wistar rats" is very unclear. I recommend the authors to revise the title to make it more clear and straightforward. Dietary supplementation with copper nanoparticles influences the markers of oxidative stress and modulates vasodilation of thoracic arteries in young Wistar rats

2. Please make a conclusion at the end of abstract. The statement "Our findings show that CuNPs possess different biological properties" is not specific enough. In conclusion, this study demonstrates that supplementation with nano Cu influences oxidative stress, which further do modify the vascular response.

3. The authors stated in the introduction what they are going to do and the reasons to test these endpoints. This is good. However, since the authors wrote two aims and different mechanisms, it is very difficult for the readers to follow all the points. I have improved the aims and mechanisms 

Therefore, I suggest the authors make a figure about the study design. This figure should also include what are the molecules to mediate the effects, and the actions of all chemicals used in this study. Figure was made

4. Please state where you bought all the chemicals, including the NPs. Also, although the authors bought Cu NPs, it is still necessary to characterize them. At least, the authors should do TEM/SEM, and measure the hydrodynamic size/zeta potential in suspensions. I looked at the authors' previous study (pharmacological reports), but they did not characterize the NPs. The nano Cu particles (40-60 nm size nanopowder, 12 m2/g) were purchased from Sky Spring Nanomaterials, Inc. (Houston, TX, US), with a purity of 99.9% on a trace metals basis, with a spherical morphology of 0.19 g/cm3 bulk density, and an 8.9 g/cm3 true density. The zeta potential of NPs was determined to be −30.3 mV (phosphate-buffered saline), for more details see Ognik et al [12].

5. The authors compared the effects of Cu NPs and CuCO3. Pleast state (1) why should the comparison be necessary it is necessary to see the difference in lipids and proteins peroxidation; We aimed to study the physiological effects of diet supplemented with copper (Cu) nanoparticles (NPs) so does comparison is necessary. 

(2) through the comparison, what conclusions could be made?

Conclusion

 Our results demonstrate that dietary nano Cu and Cu carbonate increase lipid peroxidation (TBARS). However, nano Cu supplementation has different physiological effect towards protein oxidation, which is reflected in an increase in thiols and decrease in carbonyl groups. 

Moreover, both nano Cu and carbonate enhance vasodilation to ACh, CORM-2 and A23187, meanwhile supplementation with Cu carbonate enhances vasodilation to 8-bromo-cGMP.

In addition, we provide evidence that although nano Cu may increase NO production through increased iNOS activity, the cGMP/PKG signaling cascade is downregulated, which could diminish the enhanced iNOS activation. Moreover, the 20-HETE-mediated signaling pathway is not involved in a vascular regulation of Cu supplemented rats. 

This study demonstrates that supplementation with nano Cu influences oxidative stress which modifies the response of rat thoracic arteries.

---

## [Decision Letter · Decision Letter 1]

4 Feb 2020

Dietary supplementation with copper nanoparticles influences the markers of oxidative stress and modulates vasodilation of thoracic arteries in young Wistar rats

PONE-D-19-30444R1

Dear Dr. Majewski,

We are pleased to inform you that your manuscript has been judged scientifically suitable for publication and will be formally accepted for publication once it complies with all outstanding technical requirements.

With kind regards,

Michael Bader

Academic Editor

PLOS ONE

Additional Editor Comments (optional):

Reviewers' comments:

Reviewer's Responses to Questions

**Comments to the Author**

1. If the authors have adequately addressed your comments raised in a previous round of review and you feel that this manuscript is now acceptable for publication, you may indicate that here to bypass the “Comments to the Author” section, enter your conflict of interest statement in the “Confidential to Editor” section, and submit your "Accept" recommendation.

Reviewer #2: All comments have been addressed

2. Is the manuscript technically sound, and do the data support the conclusions?

Reviewer #2: Yes

3. Has the statistical analysis been performed appropriately and rigorously? 

Reviewer #2: Yes

4. Have the authors made all data underlying the findings in their manuscript fully available?

Reviewer #2: Yes

5. Is the manuscript presented in an intelligible fashion and written in standard English?

Reviewer #2: Yes

6. Review Comments to the Author

Reviewer #2: (No Response)

7. PLOS authors have the option to publish the peer review history of their article (what does this mean?). If published, this will include your full peer review and any attached files.

Reviewer #2: Yes: Yi Cao

---

## [Editor Report · Acceptance letter]

13 Feb 2020

PONE-D-19-30444R1 

Dietary supplementation with copper nanoparticles influences the markers of oxidative stress and modulates vasodilation of thoracic arteries in young Wistar rats 

Dear Dr. Majewski:

I am pleased to inform you that your manuscript has been deemed suitable for publication in PLOS ONE. Congratulations! Your manuscript is now with our production department. 

With kind regards,

on behalf of

Prof. Michael Bader 

Academic Editor

PLOS ONE